Workshop at the 6th Symposium on Advances in Approximate Bayesian Inference (non-archival), 2024 1–12

# Uncertainty-Guided Optimization on Large Language Model Search Trees

**Julia Grosse**                                           JULIA.GROSSE@UNI-TUEBINGEN.DE
*University of Tübingen, Tübingen AI Center*

**Ruotian Wu**                                             R82WU@UWATERLOO.CA
*University of Waterloo*

**Ahmad Rashid**                                           A9RASHID@UWATERLOO.CA
*Vector Institute, University of Waterloo*

**Philipp Hennig**                                         PHILIPP.HENNIG@UNI-TUEBINGEN.DE
*University of Tübingen, Tübingen AI Center*

**Pascal Poupart**                                         PPOUPART@UWATERLOO.CA
*Vector Institute, University of Waterloo*

**Agustinus Kristiadi**                                    AGUSTINUS.KRISTIADI@VECTORINSTITUTE.AI
*Vector Institute*

## Abstract

Beam search is a standard tree search algorithm when it comes to finding sequences of maximum likelihood, for example, in the decoding processes of large language models. However, the algorithm is myopic—it does not take the whole path from the root to a leaf into account. Moreover, it is agnostic to prior knowledge available about the process: It does not consider that the objective being maximized is a likelihood and thereby has specific properties like boundedness in the unit interval. Taking a probabilistic approach, we define a Dirichlet prior over the transition probabilities and obtain a posterior distribution over the most promising paths in each iteration. These distributions are helpful to define a non-myopic Bayesian-optimization-like acquisition function that allows for a more data-efficient exploration scheme than standard beam search. We discuss how to select the prior and demonstrate in on- and off-model experiments with large language models that the method achieves as high a likelihood as beam search with much fewer node expansions, avoiding excessive (costly) LLM forward passes.

## 1. Introduction

Beam search (Koehn et al., 2003) is an optimization algorithm commonly applied to graph- and often, in particular, tree-structured problems. However, the number of possible paths that need to be considered in such settings typically grows exponentially, often exceeding the computational budget required to examine them all. This inevitably leads to *computational uncertainty* (Hennig et al., 2022): an uncertainty that *could be* fully resolved if enough compute would be available to examine all paths, but in practice is present due to the limited resources. The standard beam search algorithm, while ubiquitous—e.g., in natural language processing (NLP) for generating sentences under a large language model (LLM) and especially common in summarization and translation tasks (Vaswani et al., 2017; Zhu et al., 2019; Zhang et al., 2020; Wang et al., 2022)—completely ignores this uncertainty.

In this work, we incorporate computational uncertainty into the search process to guide it in a non-myopic fashion (it takes the future nodes until the leaf into account) and importantly, in a more data-efficient manner, akin to Bayesian optimization methods (Kushner, 1964; Močkus, 1975; Garnett, 2023). These methods are recognized for their data efficiency, not merely because they quantify uncertainty, but because they exploit the structural characteristics within that uncertainty. E.g., in continuous optimization problems, prior knowledge, such as the smoothness of a function, is often available through Gaussian processes (Rasmussen and Williams, 2005) or even (Bayesian) neural networks (Hernández-Lobato et al., 2017; Kristiadi et al., 2023).

Beam search, however, performs discrete optimization—its search space is a tree. Moreover, commonly, the values or rewards associated with each node are probabilities, bounded between 0 and 1. This assumption is predominant in many NLP applications of beam search. In this discrete setting, therefore, we will assume that rewards at a node of the search tree are the components of a Categorical distribution, and the characteristic property we aim to exploit then becomes the *concentration strength*: whether the Categorical distributions are all peaked or if some of them are rather flat, too. Intuitively, one would expect this to have a strong influence on the number of beams that need to be considered. For instance, when the distribution is peaked, it becomes less likely that other paths will overtake later on and one can be more greedy. Meanwhile, when the distribution is flat, the uncertainty about which is best token choice is higher and thereby requires more exploration and computational budget. In this work we are therefore interested in whether a probabilistic model that captures this aspect of the search space can help to decide which beams should be pursued and which can be ignored. Experiments on real-world text-generation benchmarks with GPT-2 (Radford et al., 2019) suggest that this is the case: Our method finds sentences with higher rewards than beam search with significantly fewer LLM forward passes.

## 2. Setting

Let $T = (X, E)$ be a tree with nodes $X$ and edges $E$. For each edge $e_{ij} = (x_i, x_j)$ between a parent node $x_i$ and a child $x_j$ a transition probability $c(x_i, x_j) = c_{ij}$ is defined. In our setting, we have the additional constraint that the transition probabilities for sibling nodes form a Categorical distribution, i.e. $c_{ij} \in [0, 1]$ and $\sum_{j \in \text{children}(x_i)} c_{ij} = 1$. We assume that all leaves are at depth $d$ and the branching $b$ size is the same for all nodes. Note that all of these are standard assumptions in NLP, where the LLM's outputs are softmax probabilities, sentence lengths are bounded by $d$, and each node has the same number of children (the number of tokens in the vocabulary).

For a path $\langle x_0, x_1, ..., x_n \rangle$ representing a generated text, denote with $c_{x_0 \to x_n} = \prod_{i=0}^{n-1} c_{i(i+1)}$ the product of the rewards —i.e., the probability of each token—along the path. Each node $x_i$ has an optimal value $v_i$ attached. If $x_i$ is a leaf node, the optimal value $v_i$ corresponds to $c_{x_0 \to x_i}$ and for inner nodes $x_i$ it is defined recursively as the maximum of the children's optimal values $\max_{x_c \in \text{children}(x_i)} v_c$. The goal is to find a path from the root $x_0$ to a leaf $x_l$ such that $c_{x_0 \to x_l}$ is maximized. We refer to the corresponding maximum as $v^*$.

## 3. Method

### 3.1. Probabilistic model for the transition probabilities

For a node $x_i$, let $\boldsymbol{c}_i = (c_{i1}, ..., c_{ib})$ be the vector containing the transition probabilities on the edges between $x_i$ and its $b$ children. The vectors $\boldsymbol{c}_i$ define Categorical distributions, for which a Dirichlet distribution is the common choice for a prior. For tractability, we assume that the transition probabilities are distributed independently and identically (i.i.d.) as a symmetric Dirichlet distribution with concentration parameter $\alpha > 0$, i.e., $\boldsymbol{c}_i \sim \mathrm{Dir}(\alpha)$.

The parameter $\alpha$ controls how peaked the sampled probability vectors are. In the context of LLMs, for small $\alpha$, the LLM would typically strongly favor a few tokens, whereas for large $\alpha$ the Categorical distribution would closely resemble a uniform distribution over the tokens. The symmetry of the prior implies in our context that we do not have a preference for particular tokens *a priori*. We will use this prior belief about the concentration of the LLM's softmax outputs to obtain (via a sampling-based approximation) the posterior over the optimal paths in the search tree.

### 3.2. Probabilistic model for the optimal values

The optimal value $v_{x_i}$ of a node $x_i$ factorizes as the product $c_{x_0 \to x_i}$ of the transition probabilities from the root node $x_0$ to $x_i$ and a remaining term which we refer to as $\Delta_i$ i.e., we have $v_{x_i} = c_{x_0 \to x_i} \cdot \Delta_i$. Intuitively, the term $\Delta_i$, quantifies the reward/likelihood that we get in the remaining steps from $x_i$ to a leaf node when we take all remaining decisions optimally. It can be defined by the following recurrence relation:

$$\Delta_i = \begin{cases} 1 & \text{if } x_i \text{ is a leaf} \\ \max_{x_j \in \text{children}(x_i)} \{c_{ij} \cdot \Delta_j\} & \text{otherwise.} \end{cases}$$

Due to the i.i.d. assumption above (Section 3.1), we have the joint distribution $p(c_{x_0 \to x_i}, \Delta_i) = p(c_{x_0 \to x_i}) \cdot p(\Delta_i)$. Whenever a new node $x_i$ is added to the search tree, $c_{x_0 \to x_i}$ is fully observed —its distribution is simply a Dirac delta. Therefore, to sample from the posterior $p(v_{x_i} \mid c_{x_0 \to x_i})$ over $v_{x_i}$, it is sufficient to sample from $p(\Delta_i)$ and then simply scale all samples by $c_{x_0 \to x_i}$.

We now derive the approximate sampling scheme for the distribution $p(\Delta_i)$ of the differences $\Delta_i$ that we need to generate samples from the posterior $p(v_{x_i} \mid c_{x_0 \to x_i})$. For pseudocode, see Algorithm 1 in Appendix A. It closely follows the one used in (Hennig et al., 2010; Grosse et al., 2021) with Gaussian priors. We recursively approximate the prior distribution of the $\Delta_i$'s at level $l$ with Beta distributions $\mathcal{B}_l(\Delta_i)$ as they take values in $[a, b]$ with $0 < a, b < 1$. In a bottom-up approach, we generate a set of samples

$$\{\max_j c_{nj} \mid c_n \sim \mathrm{Dir}(\alpha)\}_{n=1}^N$$

for a $\Delta_i$ at level $l = d - 1$. Using these samples, we empirically fit the parameters of the Beta distribution $\mathcal{B}_{d-1}(\Delta_i)$ via maximum likelihood (AbouRizk et al., 1994). The distributions of the $\Delta_i$ are the same for all nodes on the same level due to the i.i.d. assumption, so this has to be done only once. Note that we need this approximation since the distribution of

the maximum $\max_j c_{ij}$ has no known analytic solution. We then continue, by recursively sampling sets of the form (one per level)

$$\{\max_j c_{nj} \cdot \Delta_j \mid c_n \sim \text{Dir}(\alpha), \Delta_j \sim \mathcal{B}_{l+1}(\Delta)\}_{n=1}^N$$

for a $\Delta_i$ of the level $l$ and using it to fit the parameters of $\mathcal{B}_l(\Delta_i)$. The time complexity is $\mathcal{O}(d \cdot b \cdot N)$ for computing the approximations, i.e., it is linear in the depth and width of the tree. Note that they can be computed before the search and reused across sentences, i.e., these costs are irrelevant to the decoding process itself.[1]

### 3.3. Acquisition function

At each step of the search process, a new node is explored, and its children are added to the search tree. As described in the previous section, we generate sets of samples from the posteriors $p(v_{x_i} \mid c_{x_0 \to x_i})$ for each child $x_i$. Let $\mathcal{L}$ be the set of nodes at the current boundary of the search tree, i.e., the set of potential beams. Having access to posterior samples $\{(v_{x_i})_n\}_{n=1}^N$ for the optimal values $v_{x_i}$ in all subtrees, we estimate the probability $\mathbb{P}(v_{x_i} = v^*)$ that the subtree below $x_i$ contains the maximal value $v^*$. We do this by taking the empirical frequency with which the sampled optimal value $v_{x_i}$ of node $x_i$ was the overall maximal value across all subtrees:

$$\hat{\mathbb{P}}(v_{x_i} = v^*) = \frac{1}{N} \sum_{n=1}^N \mathbb{I}\left[(v_{x_i})_n = \max_{x_j \in \mathcal{L}}(v_{x_j})_n\right].$$

This allows us to define an acquisition function $a(x)$ according to which the next node is selected for exploration:

$$x_i = \arg\max_{x \in \mathcal{L}} a(x) = \arg\max_{x \in \mathcal{L}} \hat{\mathbb{P}}(v_x = v^*).$$

While new nodes are selected greedily, they are selected based on the beliefs over the *non-myopic* optimal values $v_x$, which take the rewards in the remaining steps into account. Moreover, exploration-exploitation is indirectly implied since $a(x)$ is constructed from the posterior's samples. These reduce the risk of getting stuck in a local optimum, in contrast to standard optimization algorithms on LLMs' search trees like beam search or greedy search.

Finally, the posterior distributions over the optimal values can not only be used for the selection of new nodes but also to monitor the progress of the optimization. E.g., one can decide to stop the search as soon as the probability $\mathbb{P}(m < v_*)$ that maximum $v_*$ is higher than the maximal value $m$ found so far drops below a confidence level $\epsilon > 0$. Algorithm 2 in Appendix A contains pseudocode for the method. In order to put an upper bound on the runtime, we introduce a hard constraint $k_{max}$ on the maximum number of nodes that can be expanded per level. This means that it is no longer possible to guarantee that a certain confidence level will be achieved, but it is still possible to estimate the confidence level that was achieved in the end.

---

1. Recall that the distribution over $\Delta_i$ is still a prior distribution; its samples only become the posterior samples for $v_{x_i}$ when combined with the observation $c_{x_0 \to x_i}$.

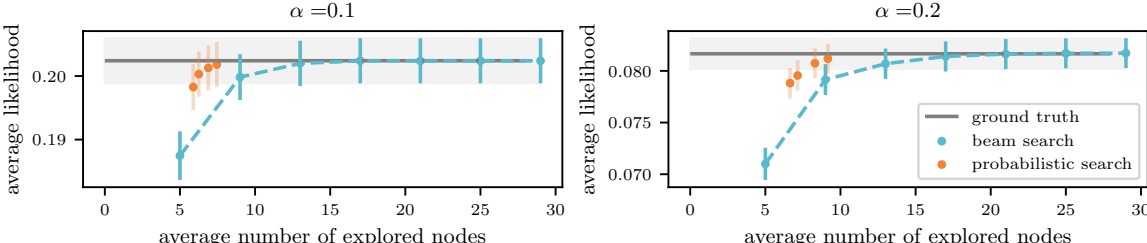

Figure 1: Comparison on trees, where the transition probabilities are sampled from a Dirichlet prior for two different values of the concentration parameter.

## 4. Experiments

### 4.1. Toy Example

As a first experiment, we compare the probabilistic search to beam search on artificially generated search problems from Dirichlet priors. The trees have branching factor $b = 8$ and depth $d = 5$. The transition probabilities at each node are sampled from a Dirichlet prior with fixed $\alpha \in \{0.1, 0.2, 0.5, 0.8\}$. The comparison is on-model, i.e., the probabilistic search is run with the ground truth parameter of $\alpha$. We repeat the experiment with different values for the confidence parameter $\epsilon$ of the probabilistic search from $\{0.05, 0.1, 0.3\}$. Since the toy problems are so small, the exploration of too many nodes is not an issue and we use $k_{max} = \infty$. Beam search is run with beam sizes ranging from 1 to 7. The results in Fig. 1 show for $\alpha = 0.1$ and $\alpha = 0.2$ that the probabilistic search dominates across the entire range of hyperparameters. The results for $\alpha = 0.5$ and $\alpha = 0.8$ (see Appendix B) show the same pattern. This suggests that knowledge of the strength of concentration helps reduce the number of search steps.

### 4.2. Experiments with LLMs

We continue with off-model experiments, where we test the probabilistic search for the decoding process of LLMs. We use GPT-2 (Radford et al., 2019) for text generation on articles from Wikipedia and the CNN Daily Mail datasets (See et al., 2017; Hermann et al., 2015). Since many of the text samples in the Wikipedia dataset end with e.g., references instead of full sentences, we filter for text samples with at least 500 tokens, resulting in a test set with 379 token sequences. We use 200 tokens as input and predict 20 tokens. We do the same for the CNN Daily Mail dataset, where we end up with 234 token sequences. We use 300 tokens as input, and predict 20 tokens. We also include a summarization task, where the goal is to generate a 40 token long summary of the input sequence. For this, we use 1000 random samples from the TL;DR dataset (Völske et al., 2017). We use the full input sequences of variable length. On the same dataset, we also experimented with a version of GPT-2 that was finetuned with human feedback for summarization tasks in particular (Stiennon et al., 2020).

Before the decoding, we fitted the concentration parameter of the Dirichlet prior based on the output of the LLM along the greedy paths on samples from the training data. It

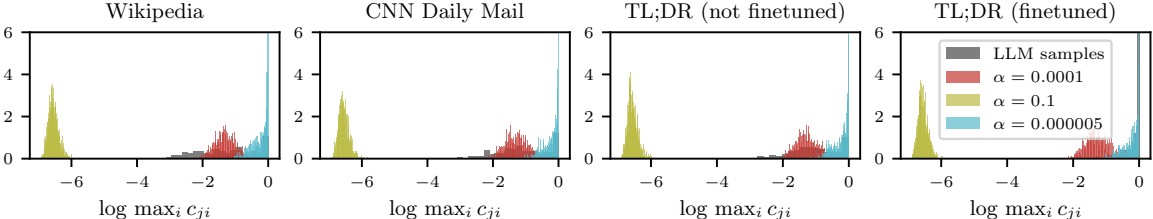

Figure 2: Distribution of the maximum of categorical distributions sampled from an LLM, as well as from Dirichlet priors with different concentrations parameters.

is possible to adjust $\alpha$ using maximum likelihood optimization (Minka, 2000). However, we have found that this did not work very well in our experiments: We suspect that it is either due to numerical problems of the optimization algorithm, or that the best fit for the Dirichlet distribution does not necessarily coincide with the best fit for the implied distribution of the (recursive) *maximum*. Figure 2 shows samples for the maximum of Categorical distributions returned by the LLM, as well as the distribution of the maximum of the Categorical distribution from a Dirichlet prior for $\alpha = 10^{-1}, 10^{-4}, 5 \times 10^{-6}$. While $\alpha = 10^{-1}$ is in the order of the maximum likelihood estimator, the other values seem to fit better. An alternative possibility is to sample the Categorical distributions used for the approximate sampling scheme in Section 3.2 not from a Dirichlet prior directly, but instead use Categorical distributions from the LLM output on sequences from the training set. To do so, we generate a set of samples by greedily decoding input sequences from the training set and tracking the observed Categorical distributions along the paths. Instead of sampling from the a Dirichlet in the sampling scheme in Section 3.2, we then draw a sample from this empirically generate set of Categorial distributions. Below, we will refer to this as "empirical prior".

Based on the results from the previous section, we run the probabilistic search with $\alpha = 10^{-4}$ (Figure 3) We use $\epsilon = 0.1$ and $k_{max} \in \{2, 3, 4, 10, 20\}$. For the beam search, we show results for $k \in \{1, 2, 3, 4, 10, 20\}$ as well. For the experiment with the fine-tuned LLM, we used $\alpha = 5 \times 10^{-6}$, $k$ and $k_{max} \in \{1, 2, 3, 4, 5\}$. No matter the choice of $k_{max}$, the probabilistic search returns on average sequences with the same or a higher log-likelihood while expanding fewer nodes (i.e., requires fewer calls to the LLM). Both ways of building the prior work well, with the empirical prior encouraging exploration a bit more.

## 5. Conclusion

We suggested a probabilistic model for the decoding process of LLM search trees. The resulting method uses the computational uncertainty over the maximum value of the optimization process along with exploiting the structure of the optimization problem to guide exploration-expoloitation in a non-myopic manner. It allows for a better search efficiency by allowing for more flexibility in the number of nodes that are expanded vis-à-vis the standard beam search. As an interesting future direction, uncertainty over the LLM's outputs (e.g. in the context of Bayesian LLMs) can be taken into account. Moreover, it is also interesting to

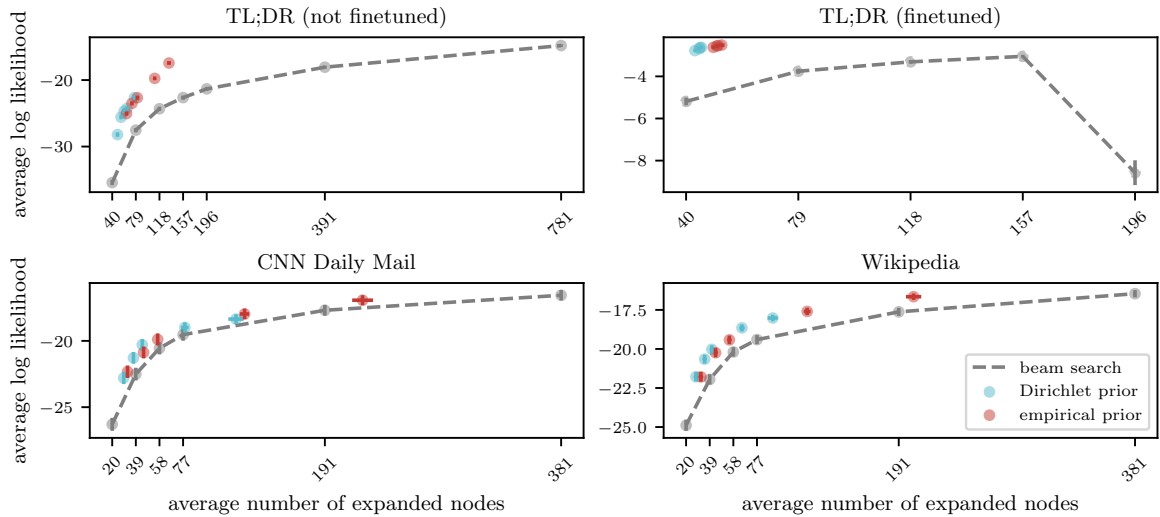

Figure 3: Comparison with beam search on LLM decoding tasks for different (maximal) beam sizes and two ways of choosing the prior.

expand the probabilistic model to incorporate (possibly uncertain) external rewards (such as ones coming from a reward model) in addition to the current LLM's likelihood rewards.

## Acknowledgments

The authors thank the International Max Planck Research School for Intelligent Systems (IMPRS-IS) for supporting JG. JG thanks Microsoft Research for support through its PhD Scholarship Programme. PH and JG gratefully acknowledge financial support by the DFG Cluster of Excellence "Machine Learning - New Perspectives for Science", EXC 2064/1, project number 390727645; the German Federal Ministry of Education and Research (BMBF) through the Tübingen AI Center (FKZ: 01IS18039A); and funds from the Ministry of Science, Research and Arts of the State of Baden-Württemberg. Resources used in this work were provided by the Province of Ontario, the Government of Canada through CIFAR, companies sponsoring the Vector Institute https://vectorinstitute.ai/partners/ and the Natural Sciences and Engineering Council of Canada. AR thanks Apple for support through the Waterloo Apple PhD Fellowship, Natural Sciences and Engineering Council of Canada for its support through the PGS-D program, and the David R. Cheriton Graduate Scholarship.

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

## Appendix A. Pseudocode

Algorithm 1 shows the pseudocode for the approximate sampling scheme described in Section 3.2 of the main text.

---

**Algorithm 1:** Computing the distributions of the $\Delta$

---

**Input:** depth of the tree $d$, branching size of tree $b$, concentration parameter $\alpha$,
      number of samples used for the approximation $N$

**Output:** table with parameters $\text{params}_l$ for beta distributions $\mathcal{B}_l$ for each level $l$ of the
      tree

**for** $l \leftarrow d - 1$ **to** $1$ **do**
    **for** $n \leftarrow 1$ **to** $N$ **do**
        $c_n \sim \text{Dirichlet}(\alpha, b)$
        **for** $j \leftarrow 1$ **to** $b$ **do**
            **if** $l = d - 1$ **then**
                $\Delta_{nj} \leftarrow 1$
            **else**
                // sample from beta distirbution with $\text{params}_{l+1}$
                $\Delta_{nj} \sim \mathcal{B}_{l+1}(\Delta)$
            **end**
        **end**
        $\Delta_n \leftarrow \max_{j=1,\ldots,b}(c_{nj} \cdot \Delta_{nj})$
    **end**
    $\text{params}_l \leftarrow \text{fit-beta-distribution}(\{\Delta_n\}_{n=1}^{N})$
**end**

---

Algorithm 2 shows the pseudocode for the full search algorithm described in Section 3.3. of the main text.

---

**Algorithm 2:** Uncertainty-guided search algorithm

---

**Input:** number of tokens to generate $d$, number of samples used for the approximation $N$, parameters for beta distributions $\mathcal{B}_l$ for all levels $l = 1, .., d$, confidence parameter $\epsilon$

**Output:** best found leaf node $x^*$

// Initialization

$\mathcal{L} \leftarrow \{x_0\}$

**for** $l \leftarrow d - 1$ **to** 1 **do**

  |   $(v_{x_0})_n \sim \mathcal{B}_1(\Delta)$ // Samples for optimal values of root $x_0$

**end**

$m \leftarrow -\infty$

$x^* \leftarrow$ None

// $v^*$ is value of overall best leaf node

**while** $\hat{\mathbb{P}}(m < v^*) > \epsilon$ **do**

  //Compute acquisition function

  **for** $x \in \mathcal{L}$ **do**

    |   $a(x) \leftarrow \frac{1}{N} \sum_{n=1}^{N} \mathbb{I}\left[(v_x)_n = \max_{x_j \in \mathcal{L}}(v_{x_j})_n\right]$

  **end**

  //Select node for expansion

  $x_i \leftarrow \arg\max_{x \in \mathcal{L}} a(x)$

  $\mathcal{L} \leftarrow (\mathcal{L} \setminus \{x_i\}) \cup children(x_i)$

  **for** $x_c \in children(x_i)$ **do**

    //Generate samples for optimal value $v_{x_c}$ of child $x_c$

    **for** $n \in 1, ..., N$ **do**

      |   $(\Delta_{x_c})_n \sim \mathcal{B}_{level(x_c)}(\Delta)$

      |   $(v_{x_c})_n \leftarrow c_{x_0 \to x_c} \cdot (\Delta_{x_c})_n$

    **end**

    //Potentially update value of best complete path so far

    **if** $level(x_c) = d$ and $c_{x_0 \to x_c}) > m$ **then**

      |   $m \leftarrow c_{x_0 \to x_c}$

      |   $x^* \leftarrow x_c$

    **end**

  **end**

  //Update estimate that optimal path was not yet found

  $\hat{\mathbb{P}}(m < v^*) \leftarrow \frac{1}{N} \sum_{n=1}^{N} \mathbb{I}\left[m \le \max_{x_j \in \mathcal{L}}(v_{x_j})_n\right]$

**end**

---

## Appendix B. Additional Experimental Results

Figure 4 shows the additional results from the experiment on on-model samples from Dirichlet priors for $\alpha = 0.5$ and $\alpha = 0.8$ as described in Section 4.1 of the main text. Here, too, the probabilistic versions outperforms beam search.

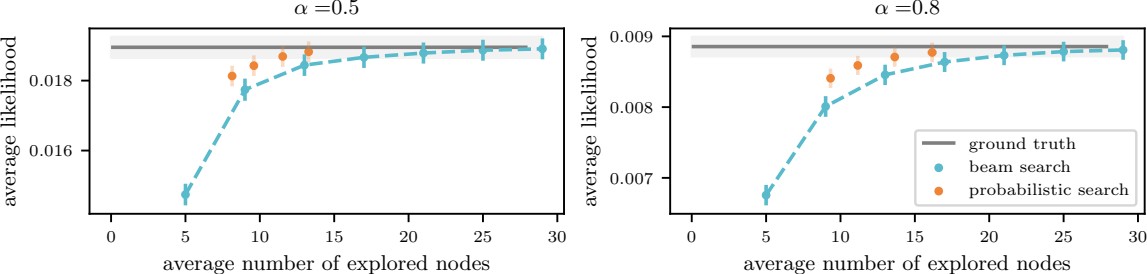

Figure 4: Comparison on trees, where the transition probabilities are sampled from a Dirichlet prior for two different values of the concentration parameter.

