# OpenReview forum: "Uncertainty-Guided Optimization on Large Language Model Search Trees"
_approximateinference.org/AABI/2024/Symposium — AABI 2024_

### Official Review · Reviewer_8jBC · 2024-04-22
**Good problem, and interesting idea**

**Rating:** 7
**Confidence:** 4

**Review:**

In the paper, the authors propose a new approach for generating high-probability sequences in large language models. The approach is a modified version of beam search over a search tree. It uses a model for the distributions of highest probability values of nodes in the tree for guiding the search. The paper describes how to learn this model from another more detailed model based on Dirichlet distribution and the training data. The experiments in the paper show promising results for two text generation tasks and one summarisation task.

The paper is well-written, and it describes an interesting and promising idea, justified partially by experiments. I support the acceptance of the paper.

When I started to read the paper, I expected that the authors' setup would include external rewards, and I got disappointed slightly when I found out that supporting those rewards was left as a future work. Saying a few more words in how to support external rewards would make reviewers like me slightly more happier. But don't misunderstand me. I am already reasonably happy with the paper, and I support its acceptance.

Here are a few minor comments.

* page 1: it practice ===> in practice
* page 3: Delta_j ~ B_{l-1}... ===> Delta_j ~ B_{l+1}
* page 4: Fianlly ===> Finally
* page 4: Algorithm 1 ===> Algorithm 2
* page 5: Some more detail on the empirical prior would be helpful for some readers like me.
* page 10: for l <- d-1 to 1 do ===> for n in 1, ..., N do
* page 10: Describing v^* explicitly would make the description of the algorithm clearer.
* page 10: c_{x0 -> xc}) ===> c_{x0 -> xc}

---

### Official Review · Reviewer_P3iJ · 2024-04-22
**Interesting new algorithm for decoding in LLMs using the structural properties of the state space but weak theory and experiment results**

**Rating:** 6
**Confidence:** 2

**Review:**

**Summary**

The paper presents a new probabilistic search algorithm that can be used for decoding in LLMs. The authors exploit the structural priors of the LLM decoding state space and transition probabilities design the new algorithm which reduces the number of total decoding steps when compared to problem-agnostic beam search while producing outputs with similar or higher log likelihood. Through experiments on a synthetic problem and GPT2, the authors show the effectiveness of the proposed method.

**Strengths**
1. The paper is well written, with careful experimental design.
2. The problem under-study is interesting and the structural properties of the problem are innovatively used to design the new algorithm.

**Weaknesses**
1. The paper proposes a new algorithm which is based on a lot of assumptions. It would have been great to analyze if these assumptions are too strong or when does they hold in practice.
2. Though the experiment design is good, the actual experiments are limited. Typically in LLM, the length is generated text is much longer than the 20 used in the paper. The authors do not provide justification for the choice. It would actually be interesting to observe the behavior as the length of generated text increases.
3. Another popular generation strategy in LLMs is to use nucleus sampling which is not based on search. I agree that it is not directly related to the topic under consideration, it would have provided a good contrast and grounding for the experiments presented.

---

### Official Review · Reviewer_jrMY · 2024-04-23
**The research presented an Uncertainty-Guided Optimization on Large Language Model Search Trees**

**Rating:** 7
**Confidence:** 4

**Review:**

The authors presented Large Language Model Search Trees using Uncertainty-Guided Optimization.
The research was experimented and the results were presented.
Some grammatical errors should be corrected to enable the semantic representation of the report to be well captured.

---

### Official Review · Reviewer_xxE8 · 2024-04-23
**Probabilistic decoding on LLM search trees**

**Rating:** 7
**Confidence:** 3

**Review:**

This paper considers a probabilistic model for the decoding process of large language model search trees. Different from standard beam search-based decoding, it maximizes an acquisition function designed by a probabilistic model for the optimal values to avoid a myopic search process. It outperforms standard beam search with respect to average likelihood values.

Strength
- This proposed model outperforms a standard beam search-based decoding process
- This model could capture uncertainties through a probabilistic manner

Weakness
- Lack of comparison with beam search with respect to time consumption
- Lack of comparison of qualitative results (such as generated text examples)
- Different evaluation metrics beyond simple likelihood
- How about using more recent LLM models instead of GPT 2

---

### Official Review · Reviewer_JxAP · 2024-04-23

**Rating:** 6
**Confidence:** 5

**Review:**

This paper presents a framework for a Bayesian-optimization-like acquisition function that accounts for priors inspired by the Beam search algorithm. To sample from $p(v_{x_i} | c_{x_0}\rightarrow x_i)$ by sampling from the likelihood $\Delta_i$ of remaining steps from $x_i$ using Beta distributions as priors.
The evaluation of this approach studies the performance compared to Beam search on synthetic data. It also studies LLM decoding (also comparing with Beam search) reported on 4 datasets with some better log-likelihood values for the proposed models. These ideas are interesting and may be worth presenting, although, the clarity of the technical contribution is less clear.

---

### Official Review · Reviewer_R7zA · 2024-04-27
**Solid piece of work on probabilistic optimisation of LLM search trees**

**Rating:** 7
**Confidence:** 2

**Review:**

The paper presents a novel approach to beam search optimization by integrating a probabilistic model that accounts for uncertainty in the search process. The approach is well-grounded in existing research, drawing on Bayesian optimization techniques to enhance data efficiency in navigating large language model (LLM) search trees. The empirical evaluation is sufficiently rigorous, including both toy examples and real-world applications with GPT-2, and it substantiates the claims of improved performance over traditional beam search methods.

The paper is generally well-written and structured. The introduction motivates the problem and the novelty of the proposed method. Sections describing the probabilistic models and the acquisition function could perhaps benefit from more detailed explanations given the density of content/ideas presented. Overall, the authors do a good job at effectively integrating established ideas of the probabilistic modeling framework into the tree search process for LLMs, resulting in an original piece of work.

The significance of this work lies in its potential to reduce computational overhead in LLM applications, which is a critical issue given the increasing size and complexity of these models. By improving the efficiency of the search process, this method could facilitate more scalable and cost-effective implementations of LLMs across various NLP tasks.

---

### Meta-Review · Area_Chair_F3LS · 2024-05-24

**Recommendation:** Accept (Poster)
**Confidence:** 4

**Metareview:**

This paper proposes a novel method for decoding LLMs inspired by BO. All of the reviewers agreed that the paper warrants acceptance. For the most part, the clarity and soundness of the paper were highlighted as strengths. The paper's topic—the intersection of probabilistic methods and LLMs—is highly relevant today and will provide an interesting point of discussion for the AABI community.

---

### Decision · Program_Chairs · 2024-05-27

Accept